# Nutritional Intake, White Matter Integrity, and Neurodevelopment in Extremely Preterm Born Infants

**DOI:** 10.3390/nu13103409

**Published:** 2021-09-27

**Authors:** Lisa M. Hortensius, Els Janson, Pauline E. van Beek, Floris Groenendaal, Nathalie H. P. Claessens, Henriette F. N. Swanenburg de Veye, Maria J. C. Eijsermans, Corine Koopman-Esseboom, Jeroen Dudink, Ruurd M. van Elburg, Manon J. N. L. Benders, Maria Luisa Tataranno, Niek E. van der Aa

**Affiliations:** 1Department of Neonatology, Wilhelmina Children’s Hospital, University Medical Center Utrecht, Utrecht University, 3584 EA Utrecht, The Netherlands; L.M.Hortensius-4@umcutrecht.nl (L.M.H.); E.Janson-6@umcutrecht.nl (E.J.); F.Groenendaal@umcutrecht.nl (F.G.); N.H.P.Claessens-2@umcutrecht.nl (N.H.P.C.); M.J.C.Eijsermans@umcutrecht.nl (M.J.C.E.); C.Koopman@umcutrecht.nl (C.K.-E.); J.Dudink@umcutrecht.nl (J.D.); rm.vanelburg@amsterdamumc.nl (R.M.v.E.); M.L.Tataranno-2@umcutrecht.nl (M.L.T.); N.vanderAa@umcutrecht.nl (N.E.v.d.A.); 2University Medical Center Utrecht Brain Center, Utrecht University, 3584 CG Utrecht, The Netherlands; 3Department of Neonatology, Máxima Medical Center, 5504 DB Veldhoven, The Netherlands; Pauline.van.Beek@mmc.nl; 4Department of Medical Psychology, Wilhelmina Children’s Hospital, University Medical Center Utrecht, Utrecht University, 3584 EA Utrecht, The Netherlands; H.deVeye@umcutrecht.nl; 5Child Development and Exercise Center, Wilhelmina Children’s Hospital, University Medical Center Utrecht, Utrecht University, 3584 EA Utrecht, The Netherlands; 6Emma Children’s Hospital, Amsterdam University Medical Center, University of Amsterdam, 1105 AZ Amsterdam, The Netherlands

**Keywords:** nutrition, extremely preterm infant, diffusion tensor imaging, white matter, neurodevelopmental outcome

## Abstract

Background: Determining optimal nutritional regimens in extremely preterm infants remains challenging. This study aimed to evaluate the effect of a new nutritional regimen and individual macronutrient intake on white matter integrity and neurodevelopmental outcome. Methods: Two retrospective cohorts of extremely preterm infants (gestational age < 28 weeks) were included. Cohort B (*n* = 79) received a new nutritional regimen, with more rapidly increased, higher protein intake compared to cohort A (*n* = 99). Individual protein, lipid, and caloric intakes were calculated for the first 28 postnatal days. Diffusion tensor imaging was performed at term-equivalent age, and cognitive and motor development were evaluated at 2 years corrected age (CA) (Bayley-III-NL) and 5.9 years chronological age (WPPSI-III-NL, MABC-2-NL). Results: Compared to cohort A, infants in cohort B had significantly higher protein intake (3.4 g/kg/day vs. 2.7 g/kg/day) and higher fractional anisotropy (FA) in several white matter tracts but lower motor scores at 2 years CA (mean (SD) 103 (12) vs. 109 (12)). Higher protein intake was associated with higher FA and lower motor scores at 2 years CA (B = −6.7, *p* = 0.001). However, motor scores at 2 years CA were still within the normal range and differences were not sustained at 5.9 years. There were no significant associations with lipid or caloric intake. Conclusion: In extremely preterm born infants, postnatal protein intake seems important for white matter development but does not necessarily improve long-term cognitive and motor development.

## 1. Introduction

Adequate nutritional intake in extremely preterm (EP) infants (gestational age (GA) < 28 weeks) is very important to achieve growth similar to fetal growth in utero. Postnatal growth is positively associated with different measures of brain development, including cortical maturation [1], linear measures of brain size [2], brain volumes [3], and neurodevelopmental outcome (NDO) [4,5,6]. Additionally, a recent systematic review showed that nutritional intake is independently associated with brain development [7]. Early cumulative fat and energy intakes were associated with larger brain volumes, improved white matter integrity [3,8], and lower brain injury scores [9]. Although a positive association between cumulative protein intake and total brain volume has been shown [8,10], only enteral protein intake has previously been associated with improved white matter integrity [3,8]. The role of the macronutrient administration route (i.e., enteral or parenteral) has not yet been fully elucidated, but enteral nutrition is suggested to contribute most to brain development [3,8].

Despite promising associations between nutrient intake and brain development, nutritional effects on NDO remain inconclusive [3,11,12,13,14,15]. A meta-analysis revealed that increased early enteral nutrition in preterm infants may support survival without neurodevelopmental impairment [16]. However, most included studies were based on outdated nutritional practices and/or reported only a short-term follow-up. Studies that assessed nutritional intake in EP infants in relation to brain development and/or NDO are often heterogeneous in methods and results, making it difficult to determine the optimal feeding regimen for this population [15,17].

Recently, we found that implementation of a new nutritional regimen with an increased protein intake (3.4 g/kg/day vs. 2.7 g/kg/day) was associated with globally increased brain volumes at 30 weeks postmenstrual age (PMA) but not at term equivalent age (TEA) [10]. In the current study, we further explored the effect of this new nutritional regimen on (a) white matter integrity at TEA and (b) NDO at 2 years CA and 5.9 years chronological age. Furthermore, we evaluated the effects of individual total, enteral, and parenteral macronutrient intakes (protein, lipids, and calories) during the first 28 postnatal days on (c) white matter integrity at TEA and (d) NDO at 2 years CA and 5.9 years chronological age. We hypothesized that the new nutritional protocol and increased macronutrient intake would be positively associated with white matter integrity and NDO.

## 2. Materials and Methods

### 2.1. Patient Population

For this retrospective study, infants from a previously derived retrospective study cohort were eligible for inclusion. The cohort consisted of 178 infants born <28 weeks GA, between January 2011 and December 2015, with nutritional parameters and segmentation of magnetic resonance imaging (MRI) at TEA available [10]. In the current study, infants were included in the white matter integrity analysis if a diffusion tensor imaging (DTI) scan at TEA of sufficient quality was available, and in the NDO analysis if neurodevelopmental assessment was performed at 2 years CA and/or 5.9 years chronological age. The medical ethics committee of the University Medical Center Utrecht approved the use of clinical and MRI data for anonymous data analysis and waived the requirement to obtain written informed consent. All data were obtained as part of the standard clinical protocol.

### 2.2. Data Collection and Nutritional Regimens

Clinical data were collected as described previously [10]. Severe brain injury was defined as grade 3 or 4 intraventricular hemorrhage according to Papile, post-hemorrhagic ventricular dilatation requiring drainage, cystic periventricular leukomalacia, and/or a cerebellar hemorrhage involving >50% of one hemisphere. Severe illness was defined as >7 days mechanical ventilation and/or abdominal surgery. Body weight Z-scores were calculated using a national reference [18]. Maternal education was defined as low, middle, or high, according to Statistics Netherlands [19]. Enteral, parenteral, and total protein (g), lipid (g), and caloric (kcal) intakes per kilogram body weight and protein/energy ratio were calculated for the first 28 postnatal days. Macronutrient content from preterm formula was sourced from nutritional information printed on the nutrition product. Energy, protein, and fat concentrations in breast milk were assumed to be 68 kcal, 1.0 g protein, and 4.0 g lipids, respectively, per 100 mL. The nutritional regimens have been described previously [10]. In summary, up to September 2013, parenteral nutrition was introduced on the second postnatal day, and protein and lipid intake were slowly increased until postnatal day 5 to a maximum of 2.6 g/kg protein and 1.7 g/kg lipids (cohort A). In September 2013, the nutritional regimen was changed to meet a new national guideline [20]. Parenteral nutrition was started as soon as possible after admission with 1 g/kg protein and lipid intake, and increased to a maximum of 3.5 g/kg protein and 4 g/kg lipids on postnatal day 3 (cohort B). In both cohorts, minimal enteral feeding was started shortly after birth and enteral intake was increased daily if tolerated, starting at 24–48 h after birth. The start and increase of enteral intake did not differ between cohort A and B.

### 2.3. MRI Acquisition and Processing

MRI was performed as previously described [10]. The DTI sequence changed in 2013, resulting in two different sequences used in this study. The previous DTI sequence used the following parameters: repetition time = 5800 ms; echo time = 70 ms; voxel size = 1.4 × 1.4 × 2.0; number of slices = 50; field of view = 18.0 cm; b value = 800 s/mm^2^; number of diffusion weighted directions = 32; number of non-diffusion weighted images = 1. The current DTI sequence used the following parameters: repetition time = 6500 ms; echo time = 80 ms; voxel size = 2.0 × 2.0 × 2.0; number of slices = 45; field of view = 16.0 cm; b value = 800 s/mm^2^; number of diffusion weighted directions = 45; number of non-diffusion weighted images = 1. The use of two DTI sequences was accounted for in the statistical analysis.

DTI data were analyzed in the source FMRIB Software Library [21]. After brain extraction and motion correction, individual FA maps were computed by fitting a tensor model to the raw diffusion data. Tract-based spatial statistics (TBSS) [22] was used to enable voxel-wise statistical analyses on the FA data. First, all individual FA maps were aligned to a representative target in common space, using the nonlinear registration tool FNIRT [23,24]. Second, a mean FA image was created and thinned to create a mean FA skeleton representing the centers of all tracts common to the group, thresholded at a FA >0.2. Third, all aligned individual FA images were projected onto this skeleton. The resulting data were used for voxel-wise statistics.

### 2.4. Neurodevelopmental Outcome Measurements

NDO was assessed at the outpatient clinic by developmental specialists (pediatric physiotherapist, child psychologist, and/or neonatologist). At 2 years CA, outcome included cognitive and (fine, gross, and total) motor development using the Bayley Scales of Infant and Toddler-Development, third edition, Dutch (Bayley-III-NL; normative mean (SD) for cognition and total motor score: 100 (15); fine and gross motor score: 10 (3)). At 5.9 years chronological age, outcome included cognition (full-scale IQ, verbal IQ, performance IQ, and processing speed), using the Wechsler Preschool and Primary Scale of Intelligence, third edition, Dutch (WPPSI-III-NL; normative mean (SD) 100 (15)) and motor development (total score, manual dexterity, aiming and catching, and balance) using the Movement Assessment Battery for Children, second edition, Dutch (M-ABC-II-NL; standard score mean (SD) 10 (3)). Higher scores indicate better functioning.

### 2.5. Statistical Analysis

#### 2.5.1. White Matter Integrity

Associations between the nutritional regimen/individual macronutrient intake and white matter integrity were evaluated. Associations between enteral/parenteral intakes and white matter integrity were analyzed post-hoc, for exploratory purposes. DTI data were further analyzed using FSL’s Randomise tool, which is a nonparametric cluster inference program using the standard general linear model [25]. Statistical inference was based on threshold-free cluster enhancement [26]. Additionally, 5000 permutations were performed to control for the family-wise error rate [27] and a threshold for statistical significance was set at *p* < 0.05. The DTI sequence (old vs. new), PMA at time of MRI, GA at birth, birth weight Z-score, sex, severe brain injury, and severe illness were included as confounders.

#### 2.5.2. Neurodevelopmental Outcome

All analyses with NDO data were performed using R version 3.5.2 [28]. Depending on the distribution of the data, Student’s *t*-tests or Mann-Whitney-U tests (continuous data) and chi-square statistics or Fisher’s Exact Test (categorical data) were performed to test for cohort differences. Cohort differences for all sub scores of the follow-up tests were analyzed post-hoc, for exploratory purposes. To further investigate the associations between nutritional regimen/daily macronutrient intake and NDO, multivariable analysis was performed, adjusting for GA at birth, birth weight Z-score, sex, severe brain injury, severe illness, and maternal education. Associations between enteral/parenteral intakes and NDO were analyzed post-hoc, for exploratory purposes. A *p*-value < 0.05 was considered significant.

## 3. Results

### 3.1. Patient Population

In total, 178 children were eligible for inclusion: 99 in cohort A and 79 in cohort B [10]. Of the total cohort, 123 infants were included in the DTI analysis (69%), and 161 (90%) and 154 (87%) children were included in follow-up analyses at 2 years CA and 5.9 years chronological age, respectively (Figure 1). Besides birth weight and birth weight Z-score, the clinical characteristics of infants with and without good-quality DTI did not differ (Appendix A Appendix A). Clinical characteristics did not differ between infants with and without follow-up at both ages (Appendix A Appendix A). Table 1 shows the baseline characteristics of the total cohort for children included in DTI and follow-up analyses at both ages. Baseline characteristics were similar in both nutritional cohorts (Appendix A Appendix A).

### 3.2. Nutritional Details

Nutritional details are shown in Table 2. Infants in cohort B had significantly higher total protein intake, higher caloric intake, and higher protein/energy ratio in the first 28 postnatal days compared to infants in cohort A. Despite the change in nutritional regimen, lipid intake was similar in cohort A and B, because infants in cohort A received more lipids than should have been provided based on the nutritional regimen at the time. Infants in cohort B received a relatively smaller amount of their proteins and lipids enterally compared to infants in cohort A.

### 3.3. Nutritional Intake and White Matter Integrity

Infants in cohort B had significantly higher FA in voxels of several white matter tracts compared to infants in cohort A. These tracts include the body of the corpus callosum, superior and posterior corona radiata, superior longitudinal fasciculus, retrolenticular part of the internal capsule, left sagittal stratum and posterior thalamic radiation, and right posterior limb of the internal capsule and superior cerebral peduncle (Figure 2A). Total protein intake during the first 28 postnatal days was significantly positively associated with FA in similar white matter tracts (Figure 2B). Cumulative enteral protein intake (g/kg) was significantly positively associated with FA in the left posterior thalamic radiation (Appendix A Appendix A), while cumulative parenteral protein intake (g/kg) was not significantly associated with FA. Total, enteral, and parenteral lipid and caloric intake were not significantly associated with FA (data not shown).

### 3.4. Nutritional Intake and Neurodevelopmental Outcome at 2 Years CA and 5.9 Years Chronological Age

Cognitive and motor outcome, including all exploratory analyzed sub scores of the follow-up tests, are shown in Table 3, and total scores are visualized in Figure 3. Total motor scores at 2 years CA were significantly lower in children in cohort B than cohort A (mean (SD) 103 (12) vs. 109 (12), respectively, *p* = 0.005), although within normal range. The difference in the total motor score seems mainly driven by significantly lower fine motor scores in children in cohort B (mean (SD) 11 (2.6) vs. 13 (2.2), *p* = 0.002). After adjustment for confounders, children in cohort B still had significantly lower total motor scores at 2 years CA (B = −5.2, 95% CI −8.9 to −1.5, *p* = 0.007) (Table 4). Cognitive scores at 2 years CA and cognitive and motor scores at 5.9 years chronological age were similar in cohort A and B, and within the normal range.

After adjusting for confounders, total protein intake was negatively associated with total motor scores at 2 years CA (B = −6.7, 95% CI −10.8 to −2.7, *p* = 0.001). Enteral and parenteral protein intake, and total, enteral, and parenteral lipid and caloric intake were not significantly associated with outcome (Table 5, Table 6 and Table 7; data on enteral and parenteral intake not shown).

## 4. Discussion

This study showed that EP infants who received the new nutritional regimen, containing more rapidly increased, higher protein intake in the first 28 postnatal days, had increased FA values in several white matter tracts at TEA but lower motor scores at 2 years CA, compared to EP infants receiving the old nutritional regimen. In addition, total and enteral protein intakes during the first 28 postnatal days were positively associated with FA at TEA in similar tracts, and total protein intake was negatively associated with motor scores at 2 years CA. Nutritional intake was not associated with NDO at 5.9 years chronological age.

### 4.1. Protein Intake and White Matter Integrity

In our study, total and enteral protein intakes during the first 28 postnatal days were positively associated with white matter integrity measured by DTI at TEA. Similar findings have previously been reported for enteral protein intake [3,8]. Enteral and total protein intakes in preterm infants have also been linked to other markers of brain development, such as linear brain measurements [29] and long-term brain connectivity [30], respectively.

Dietary protein intake is essential for brain growth and myelination during early preterm life [15,31], especially for preterm infants, as they cannot yet synthesize all non-essential amino acids endogenously [15,32,33]. Interestingly, preterm breast milk contains higher amounts of protein during the first postnatal days compared to term breast milk [34], indicating the importance of protein intake for preterm development. Protein may affect brain development through different pathways. Protein is used as building blocks in all tissues, including the brain [35,36]. In fact, 25% of myelin consists of protein [37]. Additionally, protein intake in preterm infants is positively associated with levels of insulin-like growth factor 1 (IGF-1) [38,39,40]. IGF-1 stimulates protein synthesis, glucose uptake, and cell development [33]. IGF-1 is important for brain growth in very preterm infants [41] and white matter organization in very low birth weight infants [42].

Although our findings highlight a predominant effect of total protein intake, a slightly stronger effect of enteral versus parenteral protein intake on FA was also found. In contrast to parenteral protein intake, enteral protein intake influences the gastrointestinal tract. This may reduce inflammation by elevation of IGF-1 levels [40,43,44] and by communication with the microbiome–gut–brain axis, contributing to white matter development [15,45]. However, the effect of parenteral protein intake on brain development may be confounded by illness, since the sickest infants receive parenteral nutrition for the longest time and are most vulnerable to brain injury.

### 4.2. Protein Intake and Neurodevelopmental Outcome

The negative association between total protein intake and motor development was an unexpected but not completely new finding. In preterm infants, early parenteral protein intake has been associated with lower M-ABC scores [46], higher odds on cerebral palsy [46], and lower cognitive scores [47]. Additionally, increased early total protein intake was associated with higher odds on neurodevelopmental impairments [14]. On the contrary, several studies show positive associations between total protein intake [3,12] or enteral protein intake [46,48,49] and NDO [15]. Two recent Cochrane reviews that assessed higher versus lower protein intake in formula-fed low birth weight infants (<2500 g) [50] and in parenteral nutrition in very preterm or low birth weight infants [51] could not evaluate the effect of protein intake on NDO, due to a lack of studies evaluating NDO. Conflicting and inconclusive results, together with our results, highlight the complex relationship between (the route of) protein intake and brain development.

Besides increased protein intake, infants in cohort B also had an increased protein/energy ratio compared to infants in cohort A. An adequate amount of non-protein energy is needed for protein synthesis, and to prevent unwanted oxidation of amino acids to ammonia and urea [52]. In high concentrations, ammonia and urea can be toxic [52]. Additionally, infants in cohort B received relatively more protein and lipids parenterally compared to cohort A. The aim of the new nutritional regimen was to increase protein and lipids but not to shorten the days of parenteral nutrition. Since protein and lipids are mostly provided through parenteral nutrition in the first postnatal days, and time to full enteral feeding did not change, the new nutritional regimen led to relatively more parenteral protein and lipid intake in cohort B. This may have contributed to lower motor scores in cohort B, as parenteral protein intake has previously been associated with impaired brain development and/or NDO [29,46,47].

### 4.3. Clinical Relevance

Protein intake might provide an opportunity to improve long-term outcomes through improved white matter integrity, although that could not be shown in our cohort. Global white matter development is considered a prerequisite for cognitive development [53], and FA at TEA in preterm born infants has previously been associated with improved NDO at 2 years CA [54,55].

Infants in cohort B had lower motor scores at 2 years CA compared to infants in cohort A, but the clinical relevance of this finding remains unclear. The mean cognitive and motor outcomes at 2 years CA in our cohort were well above the normative mean, even in cohort B. In addition, the differences in motor score at 2 years CA were not sustained to 5.9 years chronological age. Therefore, it seems best to conclude that providing more protein during the early postnatal period, in relatively healthy EP born infants, has no additional effect on NDO. Increased protein intake did not decrease motor scores below the normal range.

Studies that assess the effects of nutrition on long-term NDO are often underpowered, which makes it difficult to draw final conclusions. The conflicting evidence from previous studies regarding protein intake and neurodevelopment, together with our results, highlight the need for adequately powered short- and long-term follow-up studies after nutritional interventions.

### 4.4. Limitations

Our study has several limitations. First, the new nutritional regimen aimed to provide more protein and lipids, but lipid intake between cohort A and B was similar. Therefore, only protein intake differed between cohort A and B. Second, our cohort consisted of relatively healthy infants, as confirmed by the mean cognitive and motor scores within the normal range. Therefore, a potential positive effect of nutritional intake on outcome may have been more difficult to detect. Third, we have no information on breast milk or formula intake. We are aware that breast milk plays an important role in brain development [56,57]. Although we do not expect a difference in breast milk intake between cohort A and B, as there was no change in protocols regarding the use of breast milk, this cannot be completely ruled out. Fourth, the DTI sequence was changed during the study period, which might have influenced FA values. To limit the effect of the DTI sequence on our analyses, a scan protocol was included as a confounding factor in the analyses. Fifth, in 2016, there was a change in staff performing the outcome assessment. Therefore, most children in cohort B were evaluated by other clinical staff than children in cohort A. Although the new staff were equally qualified, and no differences were seen in cognitive outcome, it cannot be ruled out that this change may have contributed to the difference in motor outcome at 2 years CA. Sixth, severe brain injury was positively associated with cognitive outcome at 5.9 years chronological age. This unexpected association was likely due to the small number of infants with severe brain injury, combined with unevenly distributed levels of maternal education. By chance, all but three infants with severe brain injury had mothers with middle or high education. Seventh, our study results may have been influenced by small changes in healthcare over time due to the retrospective design of our study. However, besides a change in nutrition protocol, there were no major changes in clinical protocols. Finally, after hospital discharge, many factors may influence neurodevelopment. We only corrected our analyses for confounders from birth until TEA, and may have missed other contributing factors.

## 5. Conclusions

In our cohort of EP infants, protein intake, but not lipid or caloric intake, in the first 28 postnatal days was associated with increased FA in several white matter tracts, and lower motor scores at 2 years CA, although still within the normal range. The interplay between nutritional intake and later neurodevelopment is complex. Simply providing more protein does not necessarily contribute to improved neurodevelopment.

## Figures and Tables

**Figure 1 nutrients-13-03409-f001:**
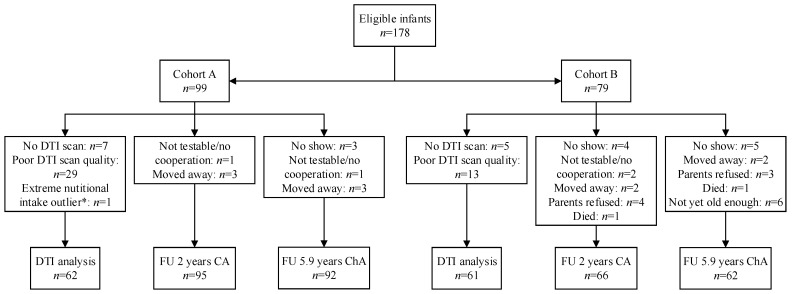
Flowchart of inclusion. DTI = diffusion tensor imaging; FU = follow-up; CA = corrected age; ChA = chronological age. * >3 standard deviations from the mean enteral and parenteral intake that caused significance.

**Figure 2 nutrients-13-03409-f002:**
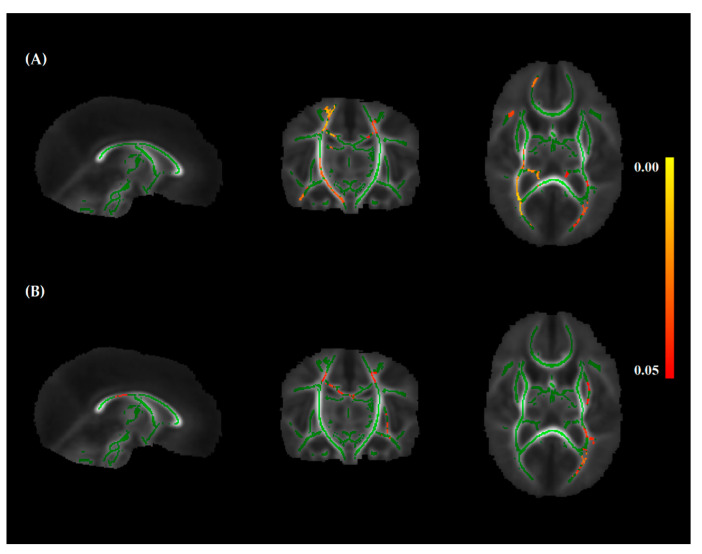
The introduction of the new nutrition protocol was associated with higher FA in several white matter tracts (**A**). Independent of the nutrition protocol, total protein intake was also associated with higher FA (**B**). Significant voxels (red-yellow; color bar indicates *p*-value) are presented on top of the mean FA skeleton (green) and are viewed from a sagittal (left), coronal (middle), and axial (right) perspective.

**Figure 3 nutrients-13-03409-f003:**
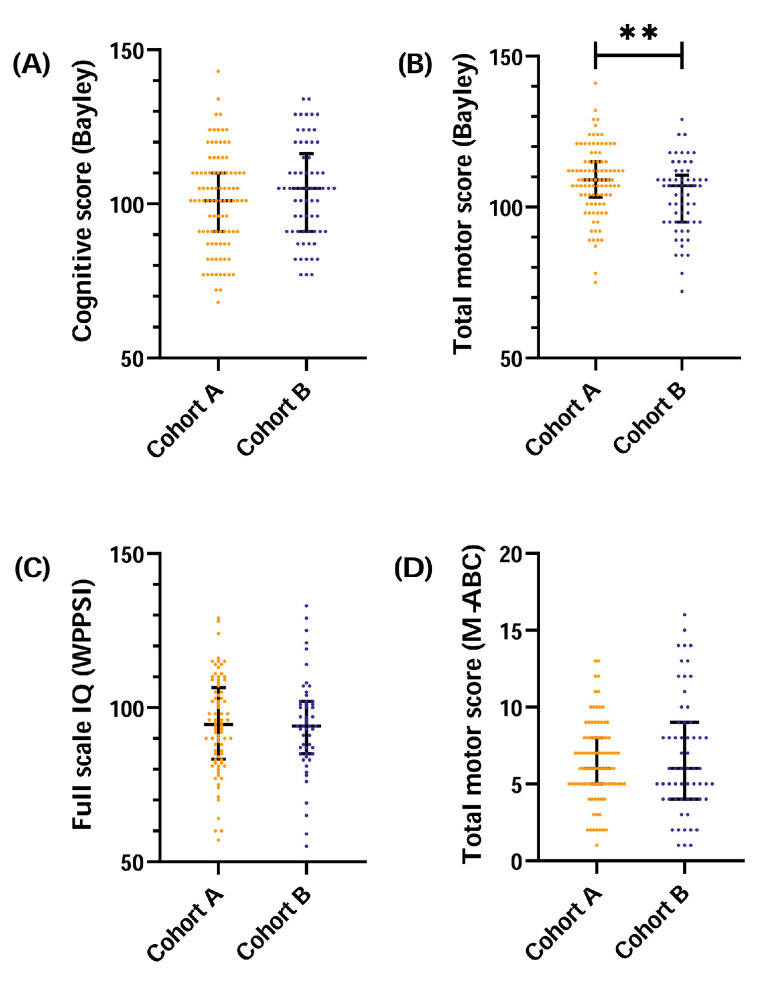
Cognitive (**A**,**C**) and motor (**B**,**D**) outcome at both follow-up ages for cohort A and B. Each data point represents a score of an individual infant. Lines represent the median with interquartile range. ** *p* < 0.01. Bayley: Bayley Scales of Infant and Toddler-Development, assessed at 2 years corrected age. WPPSI: Wechsler Preschool and Primary Scale of Intelligence, assessed at 5.9 years chronological age. M-ABC: Movement Assessment Battery for Children, assessed at 5.9 years chronological age.

**Table 1 nutrients-13-03409-t001:** Baseline characteristics of the included children.

	DTI Analysis (*n* = 123)	2 Years Corrected Age (*n* = 161)	5.9 Years Chronological Age (*n* = 154)
Male (%)	57 (46)	75 (47)	72 (47)
Gestational age (weeks) (median (Q1; Q3))	26 + 3 (25 + 6; 27 + 2)	26 + 3 (25 + 6; 27 + 1)	26 + 3 (25 + 6; 27 + 2)
Birth weight (g) (median (Q1; Q3))	880 (784; 1000)	870 (750; 995)	870 (750; 1000)
Birth weight Z-score (mean (SD))	0.39 (0.88)	0.30 (0.91)	0.28 (0.90)
SGA (<10th percentile) (%)	5 (4)	8 (5)	8 (5)
Multiplicity (%)	36 (29)	52 (32)	52 (34)
Apgar 5 min (median (Q1; Q3))	8 (7; 8)	8 (7; 9)	8 (7; 9)
Days parental nutrition (median (Q1; Q3))	13 (10; 17)	12 (10; 17)	12 (10; 17)
>7 days of ventilation (%)	62 (50)	83 (52)	79 (51)
Abdominal surgery (%)	10 (8)	14 (9)	14 (9)
Severe brain injury (%)	14 (11)	17 (11)	15 (10)
Sepsis (%)	48 (39)	62 (39)	59 (38)

DTI = diffusion tensor imaging; SGA = small for gestational age.

**Table 2 nutrients-13-03409-t002:** Nutritional details of infants in cohort A and B.

	DTI Analysis			2 Years Corrected Age			5.9 Years Chronological Age		
	Cohort A (*n* = 63)	Cohort B (*n* = 60)	*p*-Value	Cohort A (*n* = 95)	Cohort B (*n* = 66)	*p*-Value	Cohort A (*n* = 92)	Cohort B (*n* = 62)	*p*-Value
Total									
Protein (g/kg)	75 (72; 82)	97 (93; 99)	<0.001 *^#^*	74 (71; 81)	96 (93; 99)	<0.001 *^#^*	74 (71; 80)	97 (94; 99)	<0.001 *^#^*
Lipids (g/kg)	135 (121; 148)	136 (124; 144)	0.71	137 (120; 148)	134 (123; 144)	0.55	136 (119; 148)	135 (124; 144)	0.87
Calories (kcal/kg)	2924 (2710; 3157)	3075 (2895; 3181)	0.02 *	2924 (2707; 3149)	3038 (2864; 3177)	0.07	2924 (2699; 3155)	3055 (2906; 3177)	0.03 *
Daily									
Protein (g/kg)	2.7 (2.6; 2.9)	3.5 (3.3; 3.5)	<0.001 *^#^*	2.7 (2.5; 2.9)	3.4 (3.3; 3.5)	<0.001 *^#^*	2.7 (2.5; 2.8)	3.5 (3.3; 3.5)	<0.001 *^#^*
Lipids (g/kg)	4.8 (4.3; 5.3)	4.9 (4.4; 5.2)	0.71	4.9 (4.3; 5.3)	4.8 (4.4; 5.1)	0.55	4.9 (4.3; 5.3)	4.8 (4.4; 5.1)	0.87
Calories (kcal/kg)	104 (97; 113)	110 (103; 114)	0.02 *	104 (97; 112)	108 (102; 113)	0.07	104 (96; 113)	109 (104; 113)	0.03 *
% enteral									
Protein	72% (61%; 80%)	70% (59%; 76%)	0.33	73% (60%; 82%)	69% (59%; 76%)	0.05 *	72% (59%; 82%)	69% (59%; 76%)	0.12
Lipids	92% (87%; 95%)	90% (84%; 93%)	0.06	92% (87%; 95%)	90% (84%; 93%)	0.01 *	92% (87%; 95%)	90% (85%; 93%)	0.02 *
Calories	83% (75%; 89%)	83% (72%; 87%)	0.65	84% (74%; 89%)	82% (73%; 87%)	0.15	83% (74%; 89%)	82% (74%; 87%)	0.31
Protein/energy ratio									
Protein (g)/100 kcal	2.6 (2.4; 2.9)	3.2 (3.1; 3.3)	<0.001 *	2.6 (2.4; 2.9)	3.2 (3.1; 3.3)	<0.001 *	2.6 (2.4; 2.9)	3.2 (3.1; 3.3)	<0.001 *

Cumulative intake (parenteral + enteral) over first 28 postnatal days (Total), daily intake (Daily), the amount of intake received enteral (% enteral), and the protein/energy ratio (Protein/energy ratio), compared between cohort A and B. Daily intake means average daily intake. Numbers are presented as median (Q1;Q3). DTI = diffusion tensor imaging; * *p* < 0.05; ^#^
*p* < 0.001.

**Table 3 nutrients-13-03409-t003:** Cognitive and motor outcome at both follow-up ages for cohort A and B.

	Cohort A	Cohort B	*p*-Value
2 years corrected age			
Cognition	*n* = 95	*n* = 66	
Cognitive score	101 (16)	104 (16)	0.20
Motor	*n* = 94	*n* = 61	
Total motor score	109 (12)	103 (12)	0.005 **
Fine motor score	13 (2.2)	11 (2.6)	0.002 **
Gross motor score	8.1 (2.7)	7.5 (2.5)	0.12
5.9 years chronological age			
Cognition	*n* = 80	*n* = 53	
Full scale IQ	94 (15)	94 (16)	0.99
Verbal IQ	98 (18)	97 (14)	0.88
Performance IQ	96 (13)	97 (15)	0.72
Processing speed	90 (16)	88 (15)	0.46
Motor	*n* = 91	*n* = 60	
Total motor score	6.4 (2.6)	6.8 (3.9)	0.92
Manual dexterity	6.7 (2.5)	7.2 (3.3)	0.26
Aiming and catching	7.7 (2.7)	7.9 (3.7)	0.73
Balance	7.9 (2.9)	7.9 (3.3)	0.92

Scores are presented as mean (SD); IQ = intelligence quotient. ** *p* < 0.01. Motor and cognitive scores 2 years corrected age: Bayley Scales of Infant and Toddler-Development. Cognitive score 5.9 years chronological age: Wechsler Preschool and Primary Scale of Intelligence. Motor score 5.9 years chronological age: Movement Assessment Battery for Children.

**Table 4 nutrients-13-03409-t004:** Multivariable regression model of the nutritional cohort and other clinical variables in relation to cognitive and motor outcomes at both follow-up ages.

	Cognition 2 Years CA	Motor 2 Years CA	Cognition 5.9 Years ChA	Motor 5.9 Years ChA
Nutritional cohort (ref = cohort A)	2.6 (−2.3 to 7.4)	−5.2 (−8.9 to −1.5) **	0.2 (−5.5 to 5.5)	0.4 (−0.6 to 1.5)
Gestational age (days)	0.1 (−0.2 to 0.5)	0.2 (−0.1 to 0.5)	−0.1 (−0.5 to 0.3)	0.0 (−0.1 to 0.1)
Gender (ref = male)	2.6 (−2.2 to 7.3)	1.1 (−2.6 to 4.7)	1.7 (−3.5 to 6.9)	1.5 (0.5 to 2.5) **
Birth weight Z-score	2.0 (−0.7 to 4.7)	2.8 (0.8 to 4.9) **	−0.9 (−3.9 to 2.1)	0.1 (−0.5 to 0.7)
Severe illness (ref = no)	1.4 (−3.7 to 6.5)	−2.4 (−6.3 to 1.4)	0.0 (−5.6 to 5.6)	−1.3 (−2.4 to −0.2) *
Maternal education (ref = low)				
middle	3.3 (−3.2 to 9.7)	1.0 (−3.8 to 5.9)	10 (2.6 to 17.5) **	0.2 (−0.6 to 2.3)
high	11.9 (5.1 to 18.6) **	2.9 (−2.1 to 8.0)	13.4 (5.8 to 21.1) **	0.8 (−0.6 to 2.3)
Severe brain injury (ref = no)	−0.7 (−8.3 to 6.9)	−2.3 (−8.2 to 3.5)	9.1 (0.3 to 17.9) *	−0.4 (−2.1 to 1.3)

Numbers presented are beta-coefficients with 95% confidence interval. CA = corrected age; ChA = chronological age; Ref = reference. * *p* < 0.05; ** *p* < 0.01.

**Table 5 nutrients-13-03409-t005:** Multivariable regression model of daily protein intake and other clinical variables in relation to cognitive and motor outcomes at both follow-up ages.

	Cognition 2 Years CA	Motor 2 Years CA	Cognition 5.9 Years ChA	Motor 5.9 Years ChA
Daily protein intake (grams/kg)	−2.7 (−8.1 to 2.7)	−6.7 (−10.8 to −2.7) **	−1.0 (−6.9 to 4.9)	−0.6 (−1.8 to 0.5)
Gestational age (days)	0.1 (−0.2 to 0.5)	0.2 (−0.1 to 0.4)	−0.1 (−0.5 to 0.3)	0.0 (−0.1 to 0.1)
Gender (ref = male)	2.3 (−2.5 to 7.0)	1.0 (−2.6 to 4.6)	1.6 (−3.6 to 6.8)	1.4 (0.4 to 2.4) **
Birth weight Z-score	1.9 (−0.8 to 4.6)	2.7 (0.7 to 4.7) *	−0.9 (−3.9 to 2.1)	0.1 (−0.5 to 0.6)
Severe illness (ref = no)	1.3 (−3.8 to 6.5)	−3.1 (−6.9 to 0.8)	−0.2 (−5.9 to 5.5)	−1.3 (−2.4 to −0.2) *
Maternal education (ref = low)				
middle	3.8 (−2.7 to 10.3)	1.8 (−3.0 to 6.6)	10.3 (2.7 to 17.9) **	0.4 (−1.0 to 1.8)
high	12.9 (6.2 to 19.6) ^#^	2.9 (−2.1 to 7.8)	13.7 (6.0 to 21.4) **	1.1 (−0.4 to 2.5)
Severe brain injury (ref = no)	−0.8 (−8.4 to 6.9)	−2.5 (−8.3 to 3.2)	9.0 (0.2 to 17.8) *	−0.5 (−2.2 to 1.3)

Numbers presented are beta-coefficients with 95% confidence interval. CA = corrected age; ChA = chronological age; Ref = reference. * *p* < 0.05; ** *p* < 0.01, ^#^
*p* < 0.001.

**Table 6 nutrients-13-03409-t006:** Multivariable regression model of daily lipid intake and other clinical variables in relation to cognitive and motor outcomes at both follow-up ages.

	Cognition 2 Years CA	Motor 2 Years CA	Cognition 5.9 Years ChA	Motor 5.9 Years ChA
Daily lipid intake (grams/kg)	0.2 (−3.4 to 3.9)	0.1 (−2.7 to 2.9)	−0.8 (−4.7 to 3.2)	0.0 (−0.7 to 0.8)
Gestational age (days)	0.1 (−0.2 to 0.5)	0.2 (−0.1 to 0.5)	−0.1 (−0.5 to 0.3)	0.0 (−0.1 to 0.1)
Gender (ref = male)	2.5 (−2.4 to 7.3)	1.3 (−2.4 to 5.1)	1.6 (−3.6 to 6.8)	1.5 (0.5 to 2.5) **
Birth weight Z-score	2.0 (−0.7 to 4.7)	2.9 (0.8 to 5.0) **	−0.9 (−3.9 to 2.1)	0.1 (−0.5 to 0.6)
Severe illness (ref = no)	1.7 (−4.1 to 7.5)	−2.4 (−6.9 to 2.1)	−0.5 (−6.8 to 5.7)	−1.3 (−2.5 to −0.1) *
Maternal education (ref = low)				
middle	3.4 (−3.1 to 9.9)	0.9 (−4.1 to 5.9)	10.1 (2.7 to 17.6) **	0.3 (−1.1 to 1.6)
high	12.4 (5.8 to 19.1) ^#^	1.7 (−3.4 to 6.8)	13.5 (5.9 to 21.0) **	0.9 (−0.5 to 2.4)
Severe brain injury (ref = no)	−0.8 (−8.4 to 6.9)	−2.5 (−8.5 to 3.5)	9.3 (0.4 to 18.1) *	−0.4 (−2.1 to 1.3)

Numbers presented are beta-coefficients with 95% confidence interval. CA = corrected age; ChA = chronological age; Ref = reference. * *p* < 0.05; ** *p* < 0.01, ^#^
*p* < 0.001.

**Table 7 nutrients-13-03409-t007:** Multivariable regression model of daily caloric intake and other clinical variables in relation to cognitive and motor outcomes at both follow-up ages.

	Cognition 2 Years CA	Motor 2 Years CA	Cognition 5.9 Years ChA	Motor 5.9 Years ChA
Daily caloric intake (kCal/kg)	0.0 (−0.3 to 0.3)	−0.1 (−0.2 to 0.1)	−0.1 (−0.3 to 0.2)	0.0 (−0.1 to 0.1)
Gestational age (days)	0.1 (−0.2 to 0.5)	0.2 (−0.1 to 0.5)	−0.1 (−0.5 to 0.3)	0.0 (−0.1 to 0.1)
Gender (ref = male)	2.4 (−2.4 to 7.2)	1.3 (−2.5 to 5.0)	1.6 (−3.6 to 6.8)	1.5 (0.5 to 2.5) **
Birth weight Z-score	2.0 (−0.7 to 4.7)	2.9 (0.8 to 5.0) **	−0.9 (−3.9 to 2.1)	0.1 (−0.5 to 0.7)
Severe illness (ref = no)	1.5 (−4.2 to 7.3)	−3.0 (−7.5 to 1.4)	−0.8 (−7.0 to 5.4)	−1.2 (−2.5 to −0.0) *
Maternal education (ref = low)				
middle	3.4 (−3.1 to 9.9)	1.0 (−4.0 to 5.9)	10.2 (2.7 to 17.6) **	0.3 (−0.5 to 2.4)
high	12.4 (5.8 to 19.1)	1.7 (−3.4 to 6.8)	13.6 (6.0 to 21.1) **	0.9 (−0.5 to 2.4)
Severe brain injury (ref = no)	−0.7 (−8.4 to 7.0)	−2.4 (−8.4 to 3.6)	9.2 (0.4 to 18.0) *	−0.4 (−2.2 to 1.3)

Numbers presented are beta-coefficients with 95% confidence interval. CA = corrected age; ChA = chronological age; Ref = reference. * *p* < 0.05; ** *p* < 0.01.

## Data Availability

The data presented in this study are available on request from the corresponding author. The data are not publicly available due to privacy restrictions and the General Data Protection Regulation does not allow sharing pseudonymized data without informed consent.

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
