# Peer review of "Nutritional Intake, White Matter Integrity, and Neurodevelopment in Extremely Preterm Born Infants"

_nutrients, 2021, doi:10.3390/nu13103409_

Round 1
Reviewer 1 Report
The research work by Hortensius and co-workers adds important new insights on the topic of early advanced nutritional supply and psychomotor outcome until pre-school age of 5.9 years. I congratulate the authors to their pertinence to such an approach although the results do not delineate a benefit of higher protein intake in the first 4 weeks of life in preterm infants. I have some comments that can help improve the manuscript:
- The authors mention that the infants included were also taking part in another prospective study. It would be good to know as well as the ethics committee correspondence that no formal ethics vote is required. This is to secure the data as i.e. brain MRI is not routine in all countries and units.
- The authors used multivariable risk adjustment: within the magnitude of risk factors what happens to their results when important risk variables of antenatal steroids, multiple birth and sex are included.
- Breast milk is an important benefit for the studied outcome variables. What about breast milk feeding status in the two cohorts?
- What about changes in routine care in this 5 year period, particularly interventions that impact on the microbiome like probiotics or antibiotic therapy?
- Were changes in protein composition in breast milk during lactation and variations in protein content in preterm formula respected in the calculation of protein intake? What about protein quality?
- What about days on parenteral nutrition-these data would be of interest in the context of the actual recommendations to advanced enteral feeding regimes?
- The additional presentation of carbohydrates would be of interest as carbohydrates where recently associated with development in newborn infants. Furthermore, carbohydrate/protein ratio is of clinical relevance.
- The authors provide nutritional supply for a 4-week study period. It would be interesting to see whether the significant differences in total protein intake are related to differences in all 4 weeks or just a sub-period. They mention the different regimes but it would be nice to see.
- In Figure 3, annotation of statistically significant differences is missing.
- How do the authors explain the positive regression between brain injury and mental outcome at 5.9 years and the zero effect by severe illness? Where there outliers? What are the results when multiple risk adjustment for additional factors like maternal education is executed?
- In Table 4A probably the total caloric intake is presented.
- It would be good to see other morbidities of prematurity including ROP, PVL and BPD that impact on brain development and psychomotor outcome. Where there disparities between the two cohorts?
Author Response
Response to Reviewer 1 Comments
Point 1: The authors mention that the infants included were also taking part in another prospective study. It would be good to know as well as the ethics committee correspondence that no formal ethics vote is required. This is to secure the data as i.e. brain MRI is not routine in all countries and units.
Response 1: The medical ethics committee of the UMCU approved the use of clinical and MRI data for anonymous data analysis and waived the requirement to obtain written informed consent. We are able to share a copy of the METC approval upon request. The data of included infants was previously used in another retrospective and not prospective study. In addition, at the Wilhelmina Children’s Hospital in Utrecht, The Netherlands, brain MRI scanning at term-equivalent age and cognitive and motor assessment at 2 years corrected age and at 5.9 years chronological age are part of the standard clinical care. We have emphasized this in the new version of our manuscript.
Point 2: The authors used multivariable risk adjustment: within the magnitude of risk factors what happens to their results when important risk variables of antenatal steroids, multiple birth and sex are included?
Response 2: Indeed, all regression analyses on the associations between nutritional intake and neurodevelopmental outcome were adjusted for multiple risk factors, including gestational age at birth, birth weight Z-score, sex, severe brain injury, and severe illness. These risk factors were considered most important to control for, based on previous research by our and other groups. Incidence of multiple birth (i.e., multiplicity) did not differ significantly between the two cohorts (Supplemental Table 2) and was therefore not used as additional risk factor.
As requested, the multivariate regression analyses were repeated including multiple birth and antenatal steroids, one by one, as additional risk factor. Both multiplicity (0 = single, 1 = multiple birth) and use of antenatal steroids (0 = none or incomplete, 1 = complete) were not significant covariates in any of the analyses and did not change the previously significant associations between nutritional intake and motor outcome at 2 years.
Although the infants included in the follow-up analyses at 2 and 5.9 years in cohort A had significantly higher rates of antenatal steroid use compared to infants in cohort B (p = 0.006), we decided not to include this variable in our analyses. First, our aim was to study the effect of the nutritional protocol on white matter development and long-term outcome, and this association persisted when the use of antenatal steroids was added to the analysis. The effect of antenatal steroid courses on long term outcome has been previously described by many research groups. Second, we aimed to keep the risk of overfitting low by limiting the number of covariates.
Point 3: Breast milk is an important benefit for the studied outcome variables. What about breast milk feeding status in the two cohorts?
Response 3: The reviewer raises an important issue. We are very much aware of the role of breast milk in brain development, especially in preterm born infants. We therefore actively promote the use of human milk in our wards. Although we did not collect any data on the actual amount of breast milk feeding in our cohort, there was no change in breast milk feeding protocol between the two cohorts during the study period. We have further emphasized this in our limitations.
Point 4: What about changes in routine care in this 5-year period, particularly interventions that impact on the microbiome like probiotics or antibiotic therapy?
Point 5: Were changes in protein composition in breast milk during lactation and variations in protein content in preterm formula respected in the calculation of protein intake? What about protein quality?
Response 5: Due to the retrospective design of our study these data were not available. As described previously by van Beek et al. (2020), macronutrient content from preterm formula was sourced from nutritional information printed on the nutrition product. Energy, protein, and fat concentrations in breast milk were assumed to be: 68 kcal, 1.0 g protein, and 4.0 g fat, respectively, per 100 ml. This information was added to the new version of our manuscript. We do not expect protein composition to be different between our two cohorts, as our formula and breast milk policy did not change over the study period.
Point 6: What about days on parenteral nutrition-these data would be of interest in the context of the actual recommendations to advanced enteral feeding regimes?
Response 6: The duration of total parenteral nutrition did not differ between the two cohorts. We would like to refer the reviewer to Supplemental Table 2. Given the design of the study and the fact that no differences were found in the duration of total parenteral nutrition between the two cohorts, we find it difficult to provide recommendations to advanced enteral feeding regimes.
Point 7: The additional presentation of carbohydrates would be of interest as carbohydrates where recently associated with development in newborn infants. Furthermore, carbohydrate/protein ratio is of clinical relevance.
Response 7: Although we realize that carbohydrates also play an important role in preterm infants, especially when taken enterally, these data were not available in our cohort.
Point 8: The authors provide nutritional supply for a 4-week study period. It would be interesting to see whether the significant differences in total protein intake are related to differences in all 4 weeks or just a sub-period. They mention the different regimes but it would be nice to see.
Response 8: We agree that the difference in total protein intake between cohorts would be interesting to see over the weeks. Although we do not have these data available, we expect this difference to be largest during the first two weeks. The reason is two-folded: first, the new nutritional protocol does not only provide more, but especially more rapidly increased protein intake during the first 5 postnatal days compared to the old nutrition protocol. Second, infants in cohort B – receiving the new nutrition protocol – obtain significantly less protein from enteral nutrition compared to infants in cohort A (Table 2). Consequently, infants in cohort B obtain more protein from parenteral nutrition as compared to infants in cohort A. As parenteral intake is highest during the first 2 weeks (Supplemental Tables 1 and 2), we expect the difference in total protein intake to be highest during the first 1-2 weeks. Nevertheless, we cannot determine whether the significant association between protein intake and motor outcome at 2 years is linked to a specific week, as we do not have these data available.
Point 9: In Figure 3, annotation of statistically significant differences is missing.
Response 9: We would like to thank the reviewer for noticing this. We have added the annotation of a significant difference to Figure 3 in the new version of our manuscript.
Point 10: How do the authors explain the positive regression between brain injury and mental outcome at 5.9 years and the zero effect by severe illness? Where there outliers? What are the results when multiple risk adjustment for additional factors like maternal education is executed?
Response 10: The positive association brain injury between and mental outcome at 5.9 years was an unexpected finding to us as well. We further discussed this finding in our limitations section: ‘This unexpected association was likely due to the small number of infants with severe brain injury, combined with unevenly distributed levels of maternal education. By chance, all but three infants with severe brain injury had mothers with middle or high education.’
Point 11: In Table 4A probably the total caloric intake is presented.
Response 11: We would like to thank the reviewer for the indication that the meaning of Table 4 is not entirely clear. We hereby like to clarify its interpretation. Table 4 present the results of the multivariate regression analyses on the associations between nutritional intake and neurodevelopmental outcome at both follow up ages. Part A shows the associations between ‘nutritional cohort’, which has been entered as a dummy variable (0 = Cohort A, 1 = Cohort B), and cognitive and motor outcome at both follow-up ages, controlling for gestational age, gender, birthweight Z-score, severe illness, maternal education, and severe brain injury. Parts B, C, and D show the associations between daily protein, lipid, and caloric intake, respectively, and cognitive and motor outcome at both follow-up ages, controlling for the same covariates. In the new version of our manuscript, we adapted the descriptions of Table 4A-D in the Results section to clarify their meaning and interpretation.
Point 12: It would be good to see other morbidities of prematurity including ROP, PVL and BPD that impact on brain development and psychomotor outcome. Where there disparities between the two cohorts?
Response 12: On request of the reviewer, we retrospectively retrieved these data from electronical medical records. Unfortunately, data on BPD was not usable due to a high number of missing values. Only one patient was diagnosed with severe cystic PVL and already grouped to ‘severe brain injury’. The variable ROP was categorized as follows: [1] no ROP, [2] mild ROP (Stage 1 and 2), and [3] moderate-severe ROP (Stage 2 with pre-plus disease or Stage 3 to 5). As the incidence of ROP was highest in cohort A (Table 1), it seems unlikely that ROP explains the lower motor scores at 2 years in cohort B. However, these differences should be interpreted with caution due to a substantial number of missing values.
Table 1. Infants in cohort B had significantly more often moderate to severe ROP compared to infants in cohort A at all ages.
|
DTI analysis |
|
2 years corrected age |
|
5.9 years chronological age |
|
|||
|
Cohort A (N = 63) |
Cohort B (N = 60) |
p-value |
Cohort A (N=95) |
Cohort B (N=66) |
p-value |
Cohort A (N=92) |
Cohort B (N=62) |
p-value |
No ROP |
44 (74.6%) |
22 (52.4%) |
0.008* |
60 (65.9%) |
24 (53.3%) |
0.015* |
59 (67.0%) |
27 (57.4%) |
0.024* |
Mild ROP (%) |
14 (23.7%) |
13 (31.0%) |
|
29 (31.9%) |
14 (31.1%) |
|
27 (30.7%) |
13 (27.7%) |
|
Moderate-severe ROP (%) |
1 (1.7%) |
7 (16.7%) |
|
2 (2.2%) |
7 (15.6%) |
|
2 (2.3%) |
7 (14.9%) |
|
*p<0.05
Reviewer 2 Report
Lisa M. Hortensius et al submitted the manuscript (ID: nutrients-1385593, as Article), entitled “ Nutritional Intake, White Matter Integrity, and Neurodevelopment in Extremely Preterm Born Infants” to be published in Nutrients, in the section “Clinical Nutrition”.
Determining the optimal nutritional regimens in extremely preterm infants remains challenging. This study aimed to evaluate the effect of a new nutritional regimen and individual macronutrient intake on white matter integrity and neurodevelopmental outcome.
Methods: The Authors investigated two retrospective cohorts of extremely preterm infants (gestational age <28 weeks) were included: Cohort B (n=79) received a new nutritional regimen, with more rapidly increased, higher protein intake compared to cohort A (n=99). Individual protein, lipid, and caloric intakes were calculated for the first 28 postnatal days. Diffusion tensor imaging was performed at term-equivalent age, and cognitive and motor development were evaluated at 2 years corrected (Bayley-III-NL) and 5.9 years chronological age (WPPSI-III-NL, MABC-2-NL). Results: Compared to cohort A, infants in cohort B had significantly higher protein intake (3.4 g/kg/day vs 2.7 g/kg/day), higher fractional anisotropy (FA) in several white matter tracts, but lower motor scores at 2 years CA (mean[SD] 103[12] vs 109[12]). Higher protein intake was associated with higher FA and lower motor scores at 2 years CA (B=-6.7, p=0.001). However, motor scores at 2 years were still within normal range and differences were not sustained at 5.9 years. There were no significant associations with lipid or caloric intake.
The Authors concluded that in extremely preterm born infants, postnatal protein intake seems important for white matter development, but does not necessarily improve long-term cognitive and motor development.
The Paper is well written. The statistical analysis and discussion is adequate.
Author Response
Response to Reviewer 2 Comments
Point 1: The paper is well written. The statistical analysis and discussion is adequate.
Response 1: We would sincerely like to thank Reviewer 2 for these compliments.
Round 2
Reviewer 1 Report
Congratulations to the authors for providing such an important research work including a 5.9 year follow-up. I have no concerns of publication in the present form but I suggest to incorporate more details from the discussions of limitations and additional analyses from responses 2,7 and 12 into the version for publication.